



# 1 Atmospheric nitrogen deposition to terrestrial ecosystems across
# 2 Germany

Martijn Schaap[1,2], Sabine Banzhaf[2], Thomas Scheuschner[3], Markus Geupel[4], Carlijn Hendriks[1],
Richard Kranenburg[1], Hans-Dieter Nagel[3], Arjo J. Segers[1], Angela von Schlutow[3], Roy Wichink
Kruit[1,5], Peter J. H. Builtjes[1,2]
[1]TNO, Utrecht, The Netherlands
[2]Free University Berlin, Institute of Meteorology, Germany
[3]Ökodata, Berlin, Germany
[4]UBA, Dessau, Germany
[5]RIVM, Bilthoven, The Netherlands
*Correspondence to:* M. Schaap (martijn.schaap@tno.nl)
**Abstract.** Biodiversity is strongly affected by the deposition of nitrogen and sulfur on terrestrial ecosystems. In this paper
we present new quantitative estimates of the deposition of atmospheric nitrogen to ecosystems across Germany. The
methodology combines prognostic and empirical modelling to establish wet deposition fluxes and land use dependent dry
and occult deposition fluxes. On average, the nitrogen deposition in Germany was estimated to be 1057 eq ha$^{-1}$ yr$^{-1}$. The
deposition maps show considerable variability across the German territory with highest deposition on forest ecosystems in
or near the main agricultural and industrial areas. The accumulated deposition over Germany of this study is systematically
lower (27 %) than provided in earlier studies. The main reasons are an improved wet deposition estimation and the
consolidation of improved process descriptions in the LOTOS-EUROS chemistry transport model. The presented
deposition estimates show a better agreement with results obtained by integrated monitoring and deposition modelling by
EMEP than the earlier results. Through comparison of the new deposition distributions with critical load maps it is estimated
that 70% of the ecosystems in Germany receive too much nitrogen.

## 1    Introduction

Anthropogenic activities generate a tenfold more reactive nitrogen (Nr) than in the late 19th century due to increased
agricultural production and energy consumption (Galloway et al., 2003). Globally half of the annually fixed nitrogen is due
to anthropogenic activities (Fowler et al., 2013). A large part of the reactive nitrogen enters the atmosphere in the form of
ammonia ($NH_3$) through animal husbandry and fertilizer use as well as in the form of nitrogen oxides ($NO_x$) through
combustion of fossil fuels (Erisman et al., 2011). The remainder is released as nitrous oxide ($N_2O$) or as nitrate ($NO_3$) to
the soil-water compartment. In Germany about 26 % of total Nr is emitted as $NO_x$ and about 30 % as $NH_3$ (Geupel and
Frommer, 2015). Deposition of reactive nitrogen has negative impacts on biodiversity and ecosystem functioning (Sutton
et al., 2011). Especially in ecosystems adapted to nutrient poor conditions, a long term and sizeable input of reactive
nitrogen may negatively affect plant communities (Bobbink et al., 1998). Field studies have shown an inverse relationship
between reactive nitrogen deposition and species diversity (Damgaard et al., 2011). To assess the extent to which an
ecosystem is at risk the critical load concept has been developed (Hettelingh et al., 1995). Currently, it is estimated that
about half of the European ecosystems receive nitrogen in excess to the critical load (Hettelingh et al., 2013).





Major sources of oxidized nitrogen in western Europe are road transport, electricity generation, and shipping (Kuenen et al., 2014). Nitrogen oxides play a key role in atmospheric chemistry (Crutzen, 1979). Only a fraction is removed from the atmosphere close to their sources as the nitrogen oxides need to be further oxidized before they are effectively deposited (Hertel et al., 2012). Reduced nitrogen emissions in the form of ammonia are mostly associated with agriculture, though other minor sources play a role (Bouwman et al., 1997). Ammonia is emitted during and after application of fertilizer to the land, from senesces of plants, animal excretion in housing systems, during grazing and after application of manure, in food processing and fertilizer production, and as a byproduct from car exhaust equipped with a three-way catalyst (Erisman et al., 2007; Galloway et al., 2003). The atmospheric lifetime of ammonia is limited to several hours as it is effectively removed by dry and wet deposition and it readily reacts with sulfuric and nitric acid to form particulate ammonium salts (Fowler et al., 2009). In contrast to oxidized nitrogen a large proportion of reduced nitrogen is deposited relatively close to its source. Through the formation of ammonium nitrate the atmospheric cycling of reduced and oxidized nitrogen are connected (Erisman and Schaap, 2004). The particulate salts have a longer atmospheric life time providing a means of long range transport of reactive nitrogen (Hertel et al., 2012). Assessments of the exposure of sensitive ecosystems and consequent development of mitigation strategies need to take into account the different behavior among the nitrogen compounds.

The development of European mitigation strategies to reduce ecosystem exposure within the UNECE-CLRTP convention is supported by atmospheric modelling using the EMEP model (Simpson et al., 2012). This modelling system is a consensus model applied to the full European domain with a coarse resolution. In the nineties the EMEP model was used on a 125 Km resolution, which was increased to 56 Km and currently 28 Km. In Germany, and most other countries, it was recognized that this resolution does not provide sufficient detail for national assessments. Moreover, establishing the deposition distributions based on modelling alone is challenging. The nitrogen cycle is complex and chemistry transport models may show significant biases against observations (Vautard et al., 2006). One of the causes of the biases is related to the precipitation information commonly used in chemistry transport models, which is often not very accurate and does not reflect small scale variability due to orographic effects resulting in relatively poor representation of the gradients in the wet deposition flux (Simpson et al., 2006). Hence, whereas there is no alternative for modelling to establish the dry deposition, empirical approaches are often favoured for the mapping of wet deposition fluxes. A large number of monitoring sites providing precipitation chemistry exist in national and European networks (Tørseth et al., 2012; Waldner et al., 2014). In many studies wet deposition distributions are obtained through an interpolation of rain water composition and subsequent multiplication with precipitation maps (Rihm and Kurz, 2001). Finally, a specific challenge concerns the assessment of the occult deposition, which may contribute considerable inputs to ecosystems at higher altitudes (Blackwell and Driscoll, 2015). We aim to quantify the critical load exceedance for nitrogen across Germany based on a national scale mapping procedure for the individual deposition pathways.

In this study we present the methodology for the assessment of the nitrogen deposition across Germany and illustrate it with results for 2009. The methodologies to assess the wet, dry and occult deposition are presented in chapter 2. The resulting new deposition estimates as well as critical load exceedances are presented and discussed in section 3. We summarize the main findings of the study in section 4.



## 2 Methodology

### 2.1 Overall approach

To estimate the nitrogen deposition to ecosystems across the German territory as good as possible a complex procedure is followed. For pragmatic and historical reasons the assessment strategy combines empirical procedures with chemistry transport modelling results. A short overview is presented in this subsection while a more detailed description of the calculation of the different deposition pathways is given in the following subsections. Figure 1 provides an overview of this procedure including the most important input and intermediate data sets as well as data flows. As there is no large dataset of dry deposition observations we rely on chemistry transport modelling to assess the land use specific dry deposition distributions across Germany. The LOTOS-EUROS CTM is used to model the dry deposition distributions at 7x7 km$^2$ across Germany. Long range transport is incorporated by nesting the German study area into a simulation over Europe as a whole. Besides the deposition fluxes also the modelled rain water concentrations are used in the next steps of the deposition assessment. As the monitoring of wet deposition is rather straightforward, a few hundred stations provide precipitation chemistry in Germany. The density of the observations allow to perform an empirical assessment of the wet deposition flux. These data are used to correct the LOTOS-EUROS rain water concentration distribution towards the observed data using residual kriging. The resulting rain water distribution is combined with a high resolution precipitation distribution to arrive at the final wet deposition estimates. In this way a highly resolved map based on empirical data is obtained that benefits from the process knowledge incorporated in the LOTOS-EUROS model.

Currently, none of the European Eulerian chemistry transport models incorporates a parameterization of the occult deposition. For countries with only small areas of upland, this will not lead to significant underestimates in total deposition. However, for elevated locations it may be a substantial contribution to total deposition. In this study the occult deposition flux is derived by estimating the deposition flux of cloud and fog water which is combined with the pollutant concentration in the cloud water. The cloud water concentrations are deduced from the rain water concentrations. The challenge to estimate the occult deposition is to capture the variability in the cloud deposition flux which is strongly dependent on altitude, slopes and local meteorology. Therefore we use high resolution meteorological data available for Germany as a whole, i.e. 7x7 km. Note that this resolution is not able to capture high resolution variability, which means that the occult deposition reflects background values for larger regions and do not reflect the deposition at very exposed sites.

To arrive at the final result the distributions of dry, wet and occult deposition fluxes are simply added. This addition takes place on the high resolution grid of the precipitation (1x1 Km$^2$). Note that although the fluxes are provided on this high resolution the underlying fluxes are not all resolved on this resolution.

### 2.2 Chemistry transport modelling

To assess the land use specific dry deposition distributions across Germany we used the 3-D regional chemistry transport model LOTOS-EUROS (Schaap et al., 2008), which is aimed at the simulation of air pollution in the lower troposphere. The model is of intermediate complexity in the sense that the relevant processes are parameterized in such a way that the computational demands are modest enabling hour-by-hour calculations over extended periods of several years within acceptable computational time. The model is a so-called eulerian grid model, which means that the calculations for advection, vertical mixing, chemical transformations and removal by wet and dry deposition are performed on a three





dimensional grid. The LOTOS-EUROS model has a long history studying the atmospheric nitrogen and sulphur cycles.
Many scientific studies have been carried out with the LOTOS-EUROS model studying secondary inorganic aerosol
(Banzhaf et al., 2015; Erisman and Schaap, 2004; Schaap et al., 2004, 2011), sea salt (Manders et al., 2010), particulate
matter (Hendriks et al., 2013; Manders et al., 2009), ozone (Beltman et al., 2013; Curier et al., 2012), nitrogen dioxide
(Curier et al., 2014; Schaap et al., 2013) and ammonia (Hendriks et al., 2016; Van Damme et al., 2014; Wichink Kruit et
al., 2012). For details on the model we refer to these publications.
Here we outline the main features of the LOTOS-EUROS version 1.10 used in this study. The partitioning between the gas
and aerosol phase for ammonia/ammonium and nitric acid/nitrate is treated by ISORROPIA2 (Fountoukis and Nenes,
2007). Reaction of nitric acid with sea salt to form coarse sodium nitrate is included in a dynamical way. This model version
also includes a pH dependent cloud chemistry scheme (Banzhaf et al., 2012). The scheme for in- and below-cloud
scavenging of gases and particles accounts for droplet saturation (Banzhaf et al., 2012). The LOTOS-EUROS model is one
of the few chemistry transport models that uses a description of the bi-directional surface–atmosphere exchange of $NH_3$
(Wichink Kruit et al., 2012). The surface–atmosphere exchange module DEPAC is used for modelling the dry deposition
of gases (Van Zanten et al., 2010). The module in LOTOS-EUROS was expanded to include the co-deposition effect of
sulphur dioxide and ammonia. The deposition of particles is represented adapting the methodology of Zhang et al. (2001).
For a detailed analysis of the impact of including these process descriptions into LOTOS-EUROS we refer to a dedicated
sensitivity study (Banzhaf et al., 2016).
The LOTOS-EUROS model was ran for the year 2009 using ECMWF meteorological data to drive the model. Through a
one-way nesting procedure a simulation over Germany was performed on a resolution of 0.125° longitude by 0.0625°
latitude, approximately 7 by 7 $km^2$. The high resolution domain is nested in a European domain with a resolution of 0.5°
longitude by 0.25° latitude, approximately 28 by 28 $km^2$. The emissions that were fed into the LOTOS-EUROS model were
different for the two modelling domains. For the European background simulation the TNO MACC-II European emission
inventory for the year 2009 (Kuenen et al., 2014) was used. For the nest the emission data for Germany were replaced by
national data. The available national data contain sector specific emissions for the year 2005 on a regular grid with a
resolution of 1/60° longitude by 1/60° latitude (about 1.2 x 1.9 $km^2$). This emission inventory has been produced by the
Institut für Zukunftsstudien und Technologiebewertung (IZT) and the University of Stuttgart within the PAREST project
(Jörß et al., 2010). This is the most up-to-date spatially distributed inventory for Germany as a whole. Note that the emission
data were produced on county basis and that land use information was used to disaggregate the emission information to a
higher resolution. This means that the detail in the emission grids is limited, explaining why the modelling was not
performed at higher resolutions than 7x7 Km. To account for the emission situation in 2009 the PAREST emissions for
Germany were scaled on a sector basis to the officially reported emission totals for 2009 as reported in 2014 by
UNECE/CLRTAP (www.uba.de). The temporal variation of the emissions is represented by monthly, day-of-the-week and
hourly time factors that break down the annual totals for each source category (Schaap et al., 2004).
For evaluation purposes we use data from the national database maintained by UBA. This database includes data for sulphur
dioxide (N=31) and nitrogen dioxide (N=45) at rural background locations. For ammonia only very few data are available
within this database. Hence, we have conducted an effort to collect ammonia measurements from passive samplers networks
operated by different institutions across the country. For 6 networks stations with data were obtained for the years 2009-
2011. Most networks provided data for 2010 and 2011, leaving 2009 less covered. Hence, we have averaged the
concentrations over the 2009-2011 period to compare to our modelled ammonia distribution. Five stations with





concentrations far above 7 µg/m$^3$ were removed from the analysis as they were considered hot spot locations. The modelled
wet deposition fluxes were compared to observed values as presented below.

## 2.3 Wet deposition estimation

Traditionally, the assessment of wet deposition fluxes to ecosystems in Germany is performed with an empirical approach
making use of observed wet deposition fluxes at a large number of stations (Builtjes et al., 2011; Gauger et al., 2008). In
this study we derive rain water concentrations at the measurement locations and interpolate these data across Germany to
arrive at a nationwide distribution. The distribution of the concentration in rain water is then multiplied with a high
resolution precipitation map to arrive at the wet deposition estimates:
$$F_{wet} = C_{rainwater} * Precipitation\ amount$$ (Equation 1)
Datasets om precipitation chemistry from various national and regional monitoring programs in Germany were compiled
providing information for 260 sites. The national UBA network (n=11) samples on a weekly rhythm, whereas the regional
networks (n=249) may operate at a weekly, two-weekly, four-weekly or monthly basis. Unfortunately, the sampling
strategies of the regional networks are not synchronised, only allowing an assessment on annual average basis. The majority
of the wet deposition data is obtained with bulk samplers as only 40 stations are equipped with wet-only samplers. Hence,
the data from the bulk samplers that pass our quality control procedures were corrected for the dry deposition into the
funnels using species dependent correction factors (Gauger et al., 2008). As the wet deposition data are obtained from many
different sources a common quality assessment and quality control (QAQC) protocol and data selection procedure was
applied to the whole database. Following EMEP protocols (EMEP, 1996) the ion balance is calculated for all samples. In
case the net ion-charge exceeds ±20%, the measurement is rejected. To remove further outliers a statistical outlier test is
performed for the time series of each station using the Grubbs test (Grubbs, 1969). The procedure is iterative in the sense
that the procedure is repeated after identifying and removing an outlier until no outliers are found anymore, or too many
entries from the series are removed. As we log-transform the data in the interpolation scheme, the procedure is applied to
the time series of log-concentrations. All in all, most data flagged invalid are largely due to the ion balance check.
A minimum valid data coverage of 40% for a given year was required to be included in further analyses. This criterion is a
compromise between including as many stations as possible and maintaining high data quality. The 40% criterion was
established based on a pragmatic approach in which we averaged the concentration in precipitation measured at UBA
stations for 1000 random subsets of the available 52 weekly measurements for different data availabilities, i.e. 100%, 80%,
60%, 40% and 20%. As expected, the variability around the annual mean increases when data availability becomes smaller.
At 40% availability the standard deviation is around 15% of the mean concentration values for sulfate, nitrate and
ammonium, which we feel is in line with uncertainties in precipitation amounts and other concentration data.
Within this study we used a residual kriging methodology to generate the rain water concentration distribution across
Germany for 2009 (Wichink Kruit et al., 2014). Within this procedure the difference between the residual between the
observations and an a priori distribution is interpolated. The a priori distribution is the modelled average rain water
concentration from the LOTOS-EUROS model. The advantage of using LOTOS-EUROS distributions as a priori is that
we use process knowledge in the interpolation, which results in better validation statistics (Wichink Kruit et al., 2014). As
there is considerable variability between observed concentrations at stations at distances close to each other there remains
a residual between the observed and optimized distribution. Evaluations of the interpolated fields with the measured data





shows that for ammonium the differences can be as large as 25%, whereas the differences for nitrate and sulfate are much
smaller (~10%). This can be explained by the much smaller gradients across Germany observed in the rain water
concentrations for nitrate and sulfate compared to those for ammonium.
Finally, the rain water concentration is multiplied by a high resolution precipitation map for Germany (see Figure 2). This
map is derived from precipitation measurements by the German Weather Service using geostatistical approach with a linear
regression between precipitation and elevation (Herzog and Muller-Westermeier, 1998). A mean error of 8% was estimated
for the annual precipitation amounts by (Herzog and Muller-Westermeier, 1998). We validated this distribution against the
independent information on precipitation amounts from the stations with precipitation chemistry. Overall, the comparison
is very good with most annual totals within 15% of each other. The higher inaccuracy reported here could well be associated
to the host of different samplers and the sometimes long sampling periods (up to one month) used within the wet deposition
networks. Field inter-comparison of different bulk and wet-only samplers has found it difficult to estimate precipitation
volumes accurately. For instance, an accuracy better than 10% was only reached for 10–20% of the individual samples
during a comparison held in the Netherlands with samplers from 20 different countries (Erisman et al., 2003).
**2.4    Occult deposition estimation**
The occult deposition computed within this work refers to nitrogen input by orographic clouds, which is the result of
condensation processes in moist air lifted by mountains. Generally, the occult flux $F_{occult}$ is derived by the multiplication of
the deposition flux of fog water $F_{Fog}$ and the pollutant concentration in the fog water $C_{Fog}$:
$$F_{occult} = F_{fog} * C_{fog}$$                    (Equation 2)
The calculation of fog water deposition ($F_{Fog}$) follows the approach by (Katata et al., 2008, 2011). In Katata et al. (2008) a
simple linear equation for the fog deposition velocity $v_d$ based only on horizontal wind speed has been derived from
numerical experiments using a detailed multilayer land surface model that includes fog deposition onto vegetation
(SOLVEG):
$$v_d = A * U$$                    (Equation 3)
where A is the slope of $v_d$ that depends on vegetation characteristics (nondimensional), and U the horizontal wind speed
[m s$^{-1}$] above the canopy. A is calculated by:
$$A = 0.0164 * \left(\frac{LAI}{h}\right)^{-0.5}$$                    (Equation 4)
where LAI is the Leaf Area Index and h the canopy height [m]. The calculations of A using Equation 4 agreed with
observations in various cloud forests with LAI/h > 0.2 (Katata et al.,2008) and it was stated that Equation 4 can be widely
used to predict cloud water deposition on forests with LAI/h > 0.2. Using $v_d$ the flux of fog water deposition $F_{Fog}$ [kg m$^{-2}$ s$^{-1}$]
$^{1}$] is calculated using:
$$F_{Fog} = v_d * \rho * q_c = A * u * \rho * q_c$$                    (Equation 5)
where $\rho$ is the air density [kg m$^{-3}$], u and $q_c$ are the horizontal wind speed [m s$^{-1}$] and the liquid water content [kg water kg
air$^{-1}$] near the surface, respectively. The accuracy of Equation 5 in the amount of fog deposition has been validated with



data on turbulent fog flux over a coniferous forest in Germany (Klemm and Wrzesinsky, 2007) with a prediction error of
13% (Katata et al., 2011).
The meteorological input to calculate the occult deposition flux was taken from the COSMO-EU model which is the
operational NWP model of the German Weather Service (DWD). COSMO-EU was chosen as it provides the meteorological
fields over Germany on a rather high grid resolution of ca. 7x7 $km^2$. Hourly data of the meteorological fields were used to
calculate the annual fog water deposition flux based on Equation 5 with
$$F_{Fog(annual)} = \sum_t v_d(t) * \rho(t) * q_c(t) = A * \sum_t u(t) * \rho(t) * q_c(t) \qquad \text{(Equation 6)}$$
where $\rho$ is the air density [kg $m^{-3}$], $q_c$ is the liquid water content [kg water kg $air^{-1}$] at the lowest atmospheric model layer
and u the horizontal wind speed at 10 m [m $s^{-1}$]. The elevation of u may be different from that of U in Equation 3 in some
cases, but this does not cause a significant error in representative wind speed according to the logarithmic wind profile in
the surface boundary layer (Katata et al., 2011).
The approach following Katata (2008;2011) as described above is based on experimental data in forests and hence, provides
an estimation of fog water deposition on forests only. Furthermore, the input on vegetation by fog is much more relevant
for forests than for other land use categories as e.g. for grassland as the area of incidence is largest for forests when they
filter the air mass passing through including fog or clouds. Hence, available studies on the occult input on vegetation are
limited on forests and therefore fog water deposition on land use categories other than forest categories are neglected here.
The mean pollutant concentration in fog water ($C_{Fog}$) was estimated from the annual mean concentration in rainwater using
so called enrichment factors (=EF):
$$C_{Cloud} = C_{Rain} * EF \qquad \text{(Equation 7)}$$
Hereby the annual mean concentrations in rainwater per species stem from the interpolated concentration fields derived for
the calculation of the wet deposition flux. The enrichment factors for the different species were derived from a compilation
of field data from studies that provide   simultaneous observations of fog and rain water chemistry (Table 1). The
underpinning studies are provided in the supplementary material. Enrichment factors are greater than unity for all species
as within all available studies and for all species the concentration in fog water was higher than in rain water. This can be
explained by a lower dilution in fog/cloud droplets as these are smaller than rain droplets and contain less water. The
variability between the individual studies is large indicating the enrichment factors may be a large source of uncertainty.


**3     Results and discussion**
**3.1     Deposition fluxes**
The estimated average deposition fluxes for Germany in 2009 are summarized in **Error! Reference source not found.**.
The estimated total deposition of reactive nitrogen amounts to 1057 eq $ha^{-1}$ $a^{-1}$ on average across the country. Almost two
thirds (64%) of the nitrogen deposition is explained by reduced nitrogen, whereas oxidised nitrogen contributes the rest
(34%). The deposition of oxidized and reduced nitrogen show distinct patterns across the country (See Figure 3). Deposition
of reduced nitrogen maximises in the north west and in the south east of the country, basically mirroring the distribution of




animal density in Germany. For reduced nitrogen, the estimated fluxes indicate that the contributions of dry (337 eq ha$^{-1}$ a$^{-1}$
) and wet (327 eq ha$^{-1}$ a$^{-1}$) deposition are almost equal on average. However, the relative contributions show considerable
variability as in source areas for ammonia the dry deposition dominates. In more natural regions the wet deposition is about
two times more important than the dry flux. For oxidized nitrogen the deposition is highest in the Ruhr area. In addition, a
number of other large agglomerations can be recognized, such as Frankfurt, Stuttgart, and Berlin. As opposed to reduced
nitrogen, the wet deposition (248 eq ha$^{-1}$ a$^{-1}$) is more important than the dry deposition (131 eq ha$^{-1}$ a$^{-1}$) as the dry deposition
velocities of NOx are relatively small compared to those of ammonia. The oxidation of nitrogen oxides to nitric acid and
subsequent formation of particulate ammonium nitrate, especially during winter and spring, favours the long range transport
and removal through precipitation. Wet deposition fluxes for both components show (secondary) maxima in areas with
high precipitation amounts, i.e. mountainous areas like the alpine region, the Black Forest, the Erz Mountains and the Harz
Mountains. The calculated contribution of occult deposition is generally negligible at low altitudes but becomes only
important in the mentioned mountainous regions. Surprisingly, the occult deposition in the black forest is estimated to be
quite low, which is associated with relatively low values of liquid water content near the surface within the COSMO-EU
model during 2009.
The dry deposition flux is strongly dependent on land use category through surface roughness and substance properties
such as solubility or reactivity. In Table 2 the land use dependent dry deposition is listed. The comparison between land
use classes clearly illustrates that the higher roughness of the forest classes cause increased dry deposition compared to low
vegetation classes such as grasslands. The average fluxes for inland surface waters and forests are about a factor 2.5 apart.
Due to the combination of empirical results for the wet and occult deposition and modelling results for the dry deposition
it is important to assess the quality of the dry deposition estimates. This can only be done indirectly as observations for dry
deposition are hardly available. Of special interest is the consistency between the modelled and observed wet deposition
fluxes. Below, we discuss the evaluation of the LOTOS-EUROS model results in more detail.

## 3.2   Evaluation of the chemistry transport modelling

### 3.2.1   Evaluation of modelled concentrations

In Figure 5 the comparison of the modelled and observed annual average concentrations are shown. The model tends to
underestimate the observed NO2 concentrations by on average 22%. For NO$_2$ there are many stations that show a close
correspondence to observed values near the one-to-one line.  However, there are also a number of stations for which the
modelled values are about 2-4 µg m$^{-3}$ lower than those observed. Overall the gradient of NO$_2$ over the country is addressed
well. For SO$_2$ the same conclusion can be drawn, albeit that on average a small overestimation is observed. As the modelling
of all the processes including deposition occurs on an hourly time resolution it is interesting to see if the model reproduced
the seasonality and variability on observation stations. Therefore, in Figure 6 examples for the time series comparison are
shown for two stations in Germany. It can be observed that the model captures the seasonal variability in both components.
Moreover, on a short time scale many of the episodes with high concentrations are captured. Similar findings have been
reported focussing on total atmospheric NO2 columns in the Netherlands (Vlemmix et al., 2015). The major episode of
nitrogen dioxide in January is captured less well which may be due to very stable conditions in parts of Germany. As the

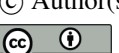



exact timing of the plumes is often off by a few hours we calculated the temporal correlation coefficient on the basis of
daily averages. The correlation coefficients ($r^2$) are very reasonable with values of 0.71 for $NO_2$ and 0.59 for $SO_2$ (see
Table 3). In short, we feel that the distributions of nitrogen dioxide and sulphur dioxide on rural background stations is
simulated satisfactorily.
Figure 5c shows the evaluation results for annual mean ammonia concentrations. On average, the model tends to
underestimate the observed concentrations slightly and yields an explained spatial variability of 65%. Hence, the model is
able to reproduce a large part of the variability and large scale gradients across Germany. Within a given region, e.g. Lower
Saxony, still considerable spread around the 1:1 line is observed, which we attribute to the low level of spatial detail in the
emission inventory within counties. Overall, the model performance for a regional assessment is promising. In a next step
is seems logical to also investigate the seasonal cycles and search for high resolution data sets. As ammonia levels are
highly variable more detailed emission information is anticipated to improve the comparison further.

### 3.2.2 Dry deposition velocity

In Table 4 the average and effective dry deposition velocities to land use classes are tabulated. The effective dry deposition
velocities defined as the annual average flux divided by the annual mean concentration are usually lower than those of the
average velocity. This is due to the anti-correlation between the dry deposition velocity and the atmospheric concentration
of most pollutants. For example, $NO_2$ concentrations show a day time and summer minimum, whereas the dry deposition
velocity maximizes at these times. Hence, the annual effective dry deposition velocity is lower than the mean of the hourly
velocities. The only exception is nitric acid because its concentration (day time and summer maximum) correlates strongly
with the dry deposition velocity leading to a higher effective than average dry deposition velocity. The distribution of the
annual mean and effective deposition velocities (at 2.5 m) show little variation across Germany although the seasonal
variability in the more continental south is larger than in the north. The deposition velocity of ammonia behaves differently
as it includes the impact of the compensation point. Figure 7a clearly illustrates the inverse relationship between the
concentration level and the effective dry deposition velocity for coniferous forest for ammonia (left panel). In the large
forest areas in Germany velocities up to 2 cm/s are modelled, whereas in ammonia rich areas in Lower Saxony and Bavaria
values below 1 cm/s are modelled. The lower dry deposition velocity in the ammonia source areas is a direct consequence
of the compensation point approach included in the dry deposition module.  In Figure 7b we compare the range of modelled
annual mean dry deposition velocities across Germany to a compilation of values reported in literature (Schrader and
Brümmer, 2014). Note that this comparison should be considered as indicative as the literature data have been obtained by
a host of different methodologies spanning different climatic conditions. Moreover, the modelled deposition velocities refer
to 2.5 m height, whereas the literature data often do not specify the representative height. Still, we conclude that the range
of the modelled dry deposition velocities for ammonia is plausible and that there are no indications that the modelled values
are unrealistic.





### 3.2.3 Wet deposition

For the evaluation of the wet deposition fluxes of LOTOS-EUROS we compare to the data of 150 stations used for the empirical assessment of the wet deposition flux. The model underestimates the wet fluxes for all components. The underestimation is lowest for reduced nitrogen (21%), see Table 5. Oxidized nitrogen shows an underestimations of 38%. In absolute terms the underestimation is about 140 eq ha$^{-1}$ yr$^{-1}$ for reactive nitrogen. In comparison to the observations the variability of the modelled wet deposition fluxes is rather small. Although models always tend to underestimate observed variability, we feel that one of the main reasons for lower variability is high spatial and temporal variability in precipitation amounts and the general challenge for meteorological models to realistically represent these variabilities. This hypothesis was tested by combining the empirically derived high resolution precipitation map and the modelled rain water concentrations. This exercise showed a considerable improvement for the spatial correlation between the calculated wet deposition fluxes. and station observations, confirming the hypothesis. It should be noted that, as expected, the exercise did not affect the bias.

### 3.3 The impact of empirical calculations

In case the underlying emissions and process knowledge is accurate the total modelled deposition using LOTOS-EUROS should be unbiased and thus highly consistent with the assessment results. Hence, deviations between the two provides hints at areas and components that need improvement in the modelling. The latter is important as a CTM is used to explore the effectivity of mitigation strategies. In Figure 8 we present the relative difference between the final assessed total deposition estimates and the modelled total deposition using LOTOS-EUROS. These ratio maps contain the signature of the highly resolved precipitation map as well as the occult deposition on top of a more general distribution. To remove the first structures it is advised to use higher resolved non-hydrostatic meteorological input data as well as to develop a parameterization for occult deposition in the chemistry transport model. The maps also clearly illustrate our finding that the model system underestimates the deposition of oxidized nitrogen. This underestimation is consistent with the air concentrations of nitrogen oxides. Moreover, this finding is consistent with a recent trend study showing that the oxidized nitrogen components are increasingly underestimated over time since 1995 (Banzhaf et al., 2015). In contrast, our model results for reduced nitrogen do not show indications for a large systematic difference as evidenced for large parts of western and central Germany. Only in the east towards the Polish border there are indications that the wet deposition is underestimated. In the southern half of Bavaria the model overestimates the wet deposition of ammonium and the assessment shows a lower total flux by about 20 %. This exceptional behaviour should be explained and we advise to investigate the emission variability as well as the precipitation statistics in more detail.

### 3.4 Comparison to previous studies

At first we compare our results  previously derived nationwide deposition maps obtained for 2007 in the MAPESI project (Builtjes et al., 2011). In principle, in MAPESI the same overall approach was taken as in this study. In comparison to MAPESI the current assessment of total deposition across Germany is lower by 27% (see Table 6). This difference is largely determined by two methodological development steps. Firstly, wet deposition QAQC criteria are more strict in this study





and the geostatistical interpolation was improved from ordinary kriging to residual kriging resulting in a 13 % lower total
deposition flux than in MAPESI (Wichink Kruit et al., 2014). Secondly, a series of model developments were consolidated
in the LOTOS-EUROS version (Banzhaf et al., 2016) The most relevant improvements were the introduction of the
compensation point for ammonia following Wichink Kruit et al. (2012), the update of the parameterization for the dry
deposition aerosols following Zhang (2001) and the introduction of a new wet deposition parameterization for below and
in-cloud scavenging following Banzhaf et al. (2012) which accounts for droplet saturation. Whereas the inclusion of these
changes hardly affects the modelled total deposition, the new process descriptions reduced the dry deposition efficiency
and led to increased wet deposition fluxes for Germany on average. The shift from dry to wet deposition reduced the bias
between modelled and observed wet deposition fluxes considerably, especially for reduced nitrogen. As the empirical
derived wet deposition maps replace the model results, this shift impacts the resulting assessment of the total deposition
across Germany. The newly modelled wet deposition fluxes by LOTOS-EUROS are closer to observations compared to
MAPESI which yields a smaller correction for the wet deposition and thus a lower total deposition estimate. Note that
within Germany the update of the model parameterizations also causes redistribution from source areas towards natural
areas leading to a smaller decline in the assessed total deposition compared to MAPESI in the large forest areas in Germany.
Hence, the reduction in comparison to MAPESI is not a homogeneous reduction across the German territory.
In Table 6 also the results of this study are compared to those of EMEP for 2009 as calculated with the emission reporting
of 2014 (www.emep.int). Our total N deposition is very close to EMEP results, with a difference of abut 6%. Altogether,
the comparison between the best estimated reduced N deposition in PINETI-2 and the reported total N deposition by EMEP
is good. The spatial distributions of the NOy and NHx deposition in the EMEP model are rather similar to ours, although
it is obvious that the distributions obtained here show much more structure than the EMEP results due to the higher
resolution modelling and high resolution precipitation distribution used here. With respect to oxidized nitrogen the final
results for this study are slightly lower than the EMEP model results. However, the LOTOS-EUROS results are significantly
lower than the results by EMEP, which is exclusively due to a difference in the wet deposition numbers of both models as
the average dry deposition fluxes are almost the same. The systematic underestimation of oxidized nitrogen in precipitation
from LOTOS-EUROS is currently under investigation.
To evaluate the total nitrogen deposition one relies on scientific studies that measure wet and dry deposition at a single site.
In Table 7 the N deposition results are compared with the estimates at few research sites in Germany. Forellenbach is an
integrated monitoring site and is located in the Southeast of Germany in the Bavarian forest. Neuglobsow is also an
integrated monitoring site and is located in the Northeast of Germany. Bourtanger Moor is a Nature2000 area that is located
in the Northwest of Germany, close to the border with the Netherlands. Note that the total N deposition at these stations
was determined using different methodologies. For Forellenbach and Neuglobsow our estimates are 20 % higher than
estimated based on the local observations. At Bourtanger Moor, a variety of methods to determine total N deposition was
explored at different locations in the nature area and a large range of total N deposition estimates was found, i.e., values
were in a range from roughly 16 till 35 kg ha$^{-1}$ yr$^{-1}$ (Mohr, 2013). Our results for Bourtanger Moor using semi-natural
vegetation is 20 Kg N ha$^{-1}$ yr$^{-1}$, which is within the observed range although slightly lower than the average of all
observations of 25 Kg N ha$^{-1}$ yr$^{-1}$. Overall, these comparisons show differences within the anticipated uncertainty as
discussed above. Unfortunately, the number of intensive monitoring stations is rather low, which highlights the need for
additional locations where dry deposition fluxes are determined.





### 3.5 Critical loads exceedance

The Critical Load concept delivers effect-based thresholds for the maximum acceptable nitrogen deposition. We compared the established deposition flux for the year 2009 to the Critical Load dataset of Germany for eutrophication (Posch et al., 2012). Regions with rather dry conditions and/or poor sandy soils appear as rather sensitive to nitrogen deposition. About 70% of the receptor area is still at risk in the year 2009 for eutrophication due to nutrient nitrogen deposition (see Figure 9). About half of the receptor area has values up to 10 kg ha$^{-1}$ a$^{-1}$ nutrient nitrogen, whereas 20 % shows even larger exceedances. Highest exceedances are found for Lower Saxony, Schleswig Holstein, North-Rhein-Westphalia, Saxony and northern Bavaria. It has to be pointed out, that the critical loads and their exceedances shown here are grid average values for a grid size of 1 km² and thus valuable for a national assessment of eutrophication or acidification only, but do not serve for local assessments. One has to bear in mind that for a certain location the recommended critical loads for such small scale or vegetation type specific assessments can differ substantially from the critical loads shown here.

### 4 Conclusions

In this study we have presented the methodology to assess the deposition of reactive nitrogen to ecosystems across Germany. The methodology combines prognostic and empirical modelling to establish land use dependent dry and occult and wet deposition fluxes. On average, the nitrogen in Germany is estimated to be 1057 eq ha$^{-1}$ yr$^{-1}$. Almost two thirds (64%) of the nitrogen deposition is explained by reduced nitrogen. Separate maps are available for the major land use classes. These maps show considerable variability across the German territory with highest deposition on forest ecosystems in or near the main agricultural and industrial areas. The results of this study are systematically lower than provided in earlier national studies, but show a better agreement with results obtained by integrated monitoring and deposition mapping by EMEP. Through comparison of the new deposition distributions with critical load maps it is estimated that 70 % of the ecosystems across Germany receive too much nitrogen.

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

Builtjes, P. and Duyzer, J.: National Implementation of the UNECE Convention on Long-range Transboundary Air
Pollution (Effects) / Nationale Umsetzung UNECELuftreinhaltekonvention (Wirkungen): Part 1: Deposition Loads:
Methods, modelling and mapping results, trends., Dessau., 2008.
Geupel, M. and Frommer, J.: Reactive Nitrogen in Germany - Causes and effects - measures and recommendations.,
Dessau., 2015.
Grubbs, F.: Procedures for Detecting Outlying Observations in Samples, Technometrics, 11, 1–21, 1969.
Hendriks, C., Kranenburg, R., Kuenen, J., van Gijlswijk, R., Wichink Kruit, R., Segers, A., Denier van der Gon, H. and
Schaap, M.: The origin of ambient particulate matter concentrations in the Netherlands, Atmos. Environ., 69, 289–303,
doi:10.1016/j.atmosenv.2012.12.017, 2013.
Hendriks, C., Kranenburg, R., Kuenen, J. J. P., Van den Bril, B., Verguts, V. and Schaap, M.: Ammonia emission time
profiles based on manure transport data improve ammonia modelling across north western Europe, Atmos. Environ., 131,
83–96, doi:10.1016/j.atmosenv.2016.01.043, 2016.
Hertel, O., Skjøth, C. a., Reis, S., Bleeker, a., Harrison, R., Cape, J. N., Fowler, D., Skiba, U., Simpson, D., Jickells, T.,
Kulmala, M., Gyldenkærne, S., Sørensen, L. L., Erisman, J. W. and Sutton, M. a.: Governing processes for reactive nitrogen
compounds in the atmosphere in relation to ecosystem, climatic and human health impacts, Biogeosciences Discuss., 9(7),
9349–9423, doi:10.5194/bgd-9-9349-2012, 2012.
Herzog, J. and Muller-Westermeier, G.: Homogenitätsprüfung und homogenisierung klimatologischer meßreihen im
deutschen wetterdienst., Deutsche Wetterdienst., 1998.
Hettelingh, J.-P., Posch, M., De Smet, P. A. M. and Downing, R. J.: The use of critical loads in emission reduction
agreements in Europe, Water, Air, & Soil Pollut., 85(4), doi:10.1007/BF01186190, 1995.
Hettelingh, J.-P., Posch, M., Velders, G. J. M., Ruyssenaars, P., Adams, M., de Leeuw, F., Lükewille, A., Maas, R., Sliggers,
J. and Slootweg, J.: Assessing interim objectives for acidification, eutrophication and ground-level ozone of the EU
National Emission Ceilings Directive with 2001 and 2012 knowledge, Atmos. Environ., 75,
doi:10.1016/j.atmosenv.2013.03.060, 2013.
Jörß, W., Kugler, U. and Theloke, J.: Emissionen im PAREST Referenzszenario 2005-2020, Dessau., 2010.
Katata, G., Nagai, H., Wrzesinsky, T., Klemm, O., Eugster, W. and Burkard, R.: Development of a land surface model
including cloud water deposition on vegetation, J. Appl. Meteorol. Climatol., 47(8), doi:10.1175/2008JAMC1758.1, 2008.



Katata, G., Kajino, M., Hiraki, T., Aikawa, M., Kobayashi, T. and Nagai, H.: A method for simple and accurate estimation
of fog deposition in a mountain forest using a meteorological model, J. Geophys. Res. Atmos., 116(20),
doi:10.1029/2010JD015552, 2011.
Klemm, O. and Wrzesinsky, T.: Fog deposition fluxes of water and ions to a mountainous site in Central Europe, Tellus,
Ser. B Chem. Phys. Meteorol., 59(4), doi:10.1111/j.1600-0889.2007.00287.x, 2007.
Kuenen, J. J. P., Visschedijk, A. J. H., Jozwicka, M. and Denier Van Der Gon, H. A. C.: TNO-MACC-II emission inventory;
A multi-year (2003-2009) consistent high-resolution European emission inventory for air quality modelling, Atmos. Chem.
Phys., 14(20), doi:10.5194/acp-14-10963-2014, 2014.
Manders, A. M. M., Schaap, M. and Hoogerbrugge, R.: Testing the capability of the chemistry transport model LOTOS-
EUROS    to    forecast    PM10    levels    in    the    Netherlands,    Atmos.    Environ.,    43(26),    4050–4059,
doi:10.1016/j.atmosenv.2009.05.006, 2009.
Manders, A. M. M., Schaap, M., Querol, X., Albert, M. F. M. A., Vercauteren, J., Kuhlbusch, T. A. J. and Hoogerbrugge,
R.:    Sea    salt    concentrations    across    the    European    continent,    Atmos.    Environ.,    44(20),    2434–2442,
doi:10.1016/j.atmosenv.2010.03.028, 2010.
Mohr, C.: Emsland: Erfassung der Stickstoffbelastungen aus der Tierhaltung zur Erarbeitung innovativer Lösungsansätze
für eine zukunftsfähige Landwirtschaft bei gleichzeitigem Schutz der sensiblen Moorlandschaft (ERNST), Emsland., 2013.
Posch, M., Slootweg, J. and Hettelingh, J.-P.: Modelling and Mapping of Atmospherically-induced Ecosystem Impacts in
Europe., 2012.
Rihm, B. and Kurz, D.: Deposition and critical loads of nitrogen in Switzerland, Water. Air. Soil Pollut., 130(1–4 III),
doi:10.1023/A:1013972915946, 2001.
Schaap, M., van Loon, M., ten Brink, H. M., Dentener, F. J. and Builtjes, P. J. H.: Secondary inorganic aerosol simulations
for Europe with special attention to nitrate, Atmos. Chem. Phys., 4(3), 857–874 [online] Available from:
http://www.scopus.com/inward/record.url?eid=2-s2.0-3242875516&partnerID=tZOtx3y1, 2004.
Schaap, M., Timmermans, R. M. A., Roemer, M., Boersen, G. A. C., Builtjes, P. J. H., Sauter, F. J., Velders, G. J. M. and
Beck, J. P.: The LOTOS EUROS model: description, validation and latest developments, Int. J. Environ. Pollut., 32(2),
270, doi:10.1504/IJEP.2008.017106, 2008.
Schaap, M., Otjes, R. P. and Weijers, E. P.: Illustrating the benefit of using hourly monitoring data on secondary inorganic
aerosol and its precursors for model evaluation, Atmos. Chem. Phys., 11(21), 11041–11053, doi:10.5194/acp-11-11041-
533    2011, 2011.

Schaap, M., Kranenburg, R., Curier, L., Jozwicka, M., Dammers, E. and Timmermans, R.: Assessing the Sensitivity of the
OMI-NO2 Product to Emission Changes across Europe, Remote Sens., 5(9), 4187–4208, doi:10.3390/rs5094187, 2013.
Schrader, F. and Brümmer, C.: Genfer Luftreinhaltekonvention der UNECE: Literaturstudie zu Messungen der Ammoniak-
Depositionsgeschwindigkeit, UBA-Texte, (67/2014), 2014.
Simpson, D., Butterbach-Bahl, K., Fagerli, H., Kesik, M., Skiba, U. and Tang, S.: Deposition and emissions of reactive
nitrogen over European forests: A modelling study, Atmos. Environ., 40(29), doi:10.1016/j.atmosenv.2006.04.063, 2006.



Simpson, D., Benedictow, A., Berge, H., Bergström, R., Emberson, L. D., Fagerli, H., Flechard, C. R., Hayman, G. D.,
Gauss, M., Jonson, J. E., Jenkin, M. E., Nyúri, A., Richter, C., Semeena, V. S., Tsyro, S., Tuovinen, J.-P., Valdebenito, A.
and Wind, P.: The EMEP MSC-W chemical transport model &ndash; Technical description, Atmos. Chem. Phys.,
12(16), doi:10.5194/acp-12-7825-2012, 2012.
Sutton, M. a, Oenema, O., Erisman, J. W., Leip, A., van Grinsven, H. and Winiwarter, W.: Too much of a good thing.,
Nature, 472(7342), 159–161, doi:10.1038/472159a, 2011.
Tørseth, K., Aas, W., Breivik, K., Fjeraa, A. M., Fiebig, M., Hjellbrekke, A. G., Lund Myhre, C., Solberg, S. and Yttri, K.
E.: Introduction to the European Monitoring and Evaluation Programme (EMEP) and observed atmospheric composition
change during 1972-2009, Atmos. Chem. Phys., 12(12), doi:10.5194/acp-12-5447-2012, 2012.
Van Damme, M., Wichink Kruit, R. J., Schaap, M., Clarisse, L., Clerbaux, C., Coheur, P.-F., Dammers, E., Dolman, A. J.
and Erisman, J. W.: Evaluating 4 years of atmospheric ammonia (NH 3 ) over Europe using IASI satellite observations and
LOTOS-EUROS model results, J. Geophys. Res. Atmos., 119(15), 9549–9566, doi:10.1002/2014JD021911, 2014.
Vautard, R., Van Loon, M., Schaap, M., Bergström, R., Bessagnet, B., Brandt, J., Builtjes, P. J. H., Christensen, J. H.,
Cuvelier, C., Graff, A., Jonson, J. E., Krol, M., Langner, J., Roberts, P., Rouil, L., Stern, R., Tarrasón, L., Thunis, P.,
Vignati, E., White, L. and Wind, P.: Is regional air quality model diversity representative of uncertainty for ozone
simulation?, Geophys. Res. Lett., 33(24), L24818, doi:10.1029/2006GL027610, 2006.
Vlemmix, T., Eskes, H. J., Piters, A. J. M., Schaap, M., Sauter, F. J., Kelder, H. and Levelt, P. F.: MAX-DOAS tropospheric
nitrogen dioxide column measurements compared with the Lotos-Euros air quality model, Atmos. Chem. Phys., 15(3),
1313–1330, doi:10.5194/acp-15-1313-2015, 2015.
Waldner, P., Marchetto, A., Thimonier, A., Schmitt, M., Rogora, M., Granke, O., Mues, V., Hansen, K., Pihl Karlsson, G.,
Žlindra, D., Clarke, N., Verstraeten, A., Lazdins, A., Schimming, C., Iacoban, C., Lindroos, A.-J., Vanguelova, E., Benham,
S., Meesenburg, H., Nicolas, M., Kowalska, A., Apuhtin, V., Napa, U., Lachmanová, Z., Kristoefel, F., Bleeker, A.,
Ingerslev, M., Vesterdal, L., Molina, J., Fischer, U., Seidling, W., Jonard, M., O'Dea, P., Johnson, J., Fischer, R. and
Lorenz, M.: Detection of temporal trends in atmospheric deposition of inorganic nitrogen and sulphate to forests in Europe,
Atmos. Environ., 95, doi:10.1016/j.atmosenv.2014.06.054, 2014.
Wichink Kruit, R., Schaap, M., Segers, A., Banzhaf, S., Scheuschner, T., Builtjes, P. and Heslinga, D.: PINETI (Pollutant
INput and EcosysTem Impact) report. Modelling and mapping of atmospheric nitrogen and sulphur deposition and critical
loads for ecosystem specific assessment of threats to biodiversity in Germany, Dessau., 2014.
Wichink Kruit, R. J., Schaap, M., Sauter, F. J., van Zanten, M. C. and van Pul, W. A. J.: Modeling the distribution of
ammonia across Europe including bi-directional surface–atmosphere exchange, Biogeosciences, 9(12), 5261–5277,
doi:10.5194/bg-9-5261-2012, 2012.
Van Zanten, M., Sauter, F., Wichink Kruit, R., Van Jaarsveld, J. and Van Pul, A.: Description of the DEPAC module: Dry
deposition modelling with DEPAC, Bilthoven., 2010.
Zhang, L., Gong, S., Padro, J. and Barrie, L.: A size-segregated particle dry deposition scheme for an atmospheric aerosol
module, Atmos. Environ., 35(3), doi:10.1016/S1352-2310(00)00326-5, 2001.



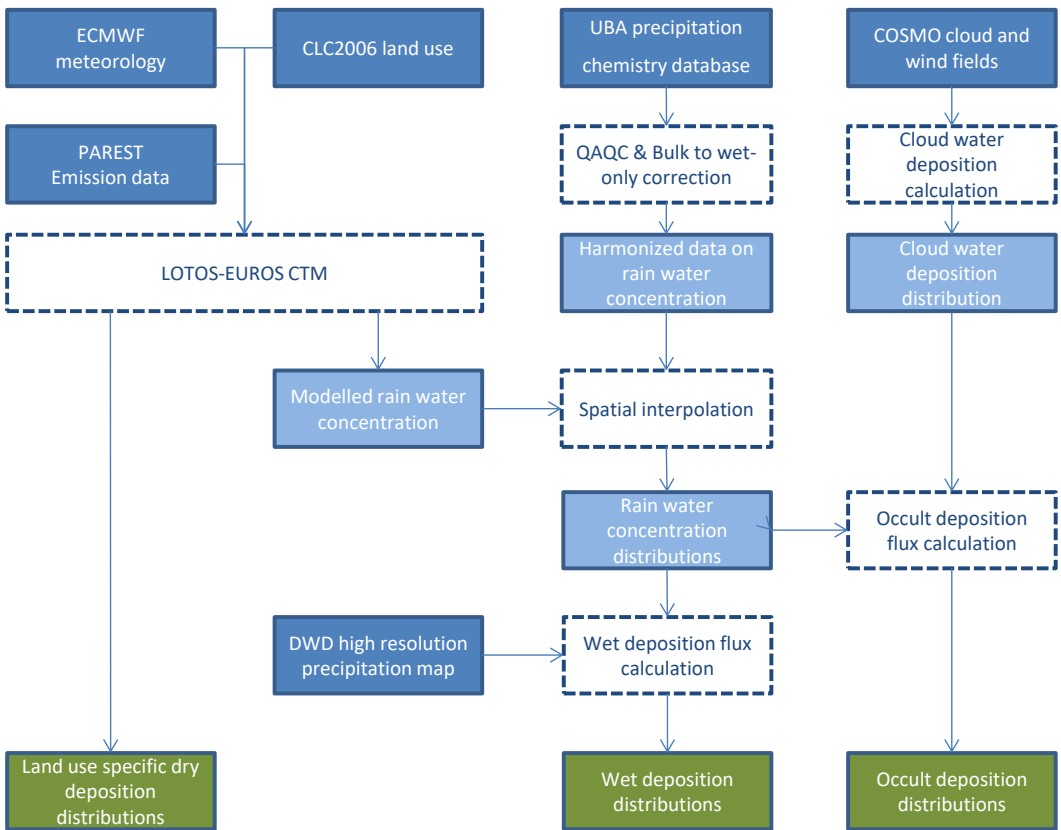


**Figure 1. Overview of the assessment methodology used in this study. The scheme introduces important input data (dark blue boxes), key intermediate results (light blue boxes), calculation steps (dashed boxes) and final results (green boxes). The arrows indicate the data flow and dependencies.**







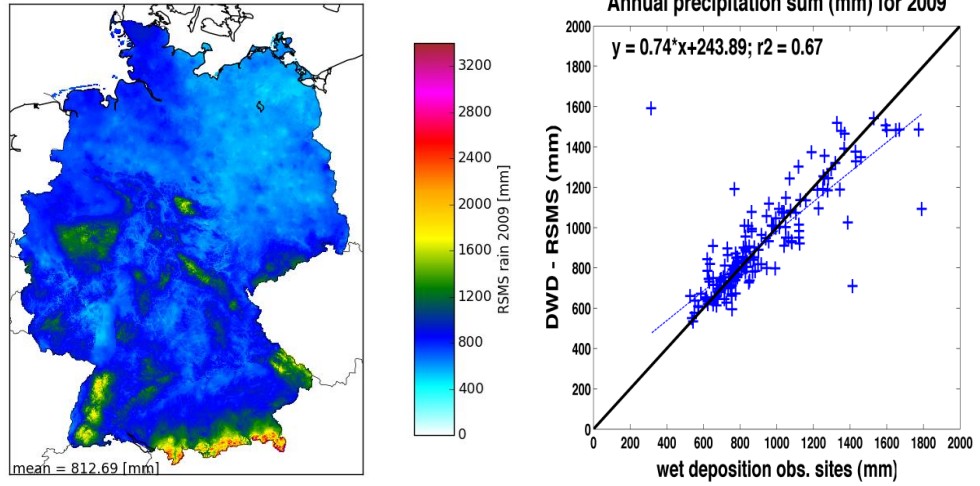

**Figure 2. High resolution precipitation map (left) and its validation against the independent data from the stations**
**with precipitation chemistry**

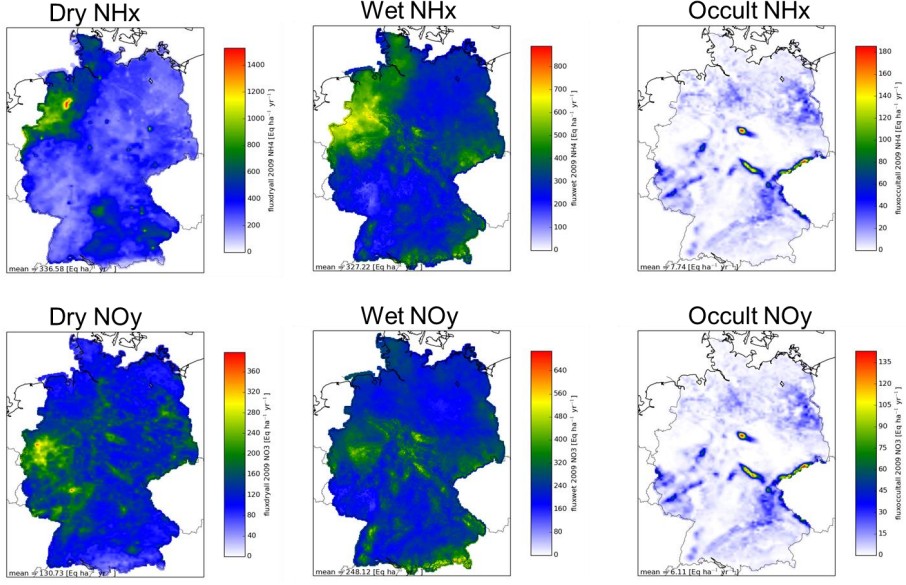


**Figure 3. Annual distributions of dry (left), wet (middle) and occult (right) deposition flux (eq ha$^{-1}$ a$^{-1}$) for reduced (top) and**
**oxidized (bottom) nitrogen for 2009.**






**Figure 4. Annual distributions of dry (top), wet (middle) and occult (bottom) deposition flux (eq ha⁻¹ a⁻¹) for reduced (left) and oxidized (right) nitrogen for 2009.**






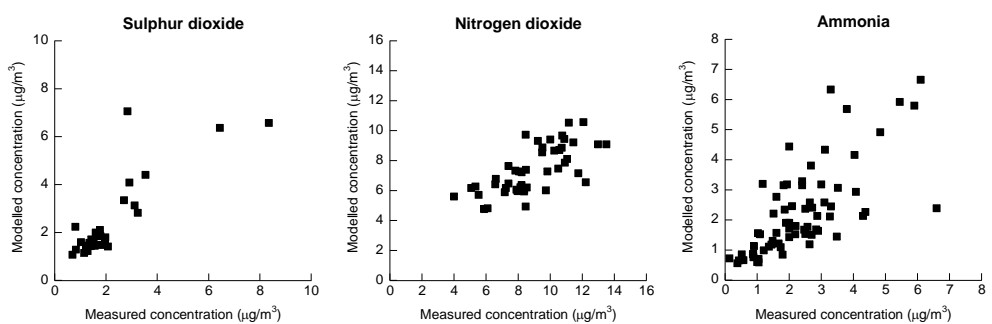


**Figure 5. Comparison between modelled and measured annual mean concentrations (µg/m³) of sulfur dioxide, nitrogen dioxide**
**and ammonia at stations across Germany**



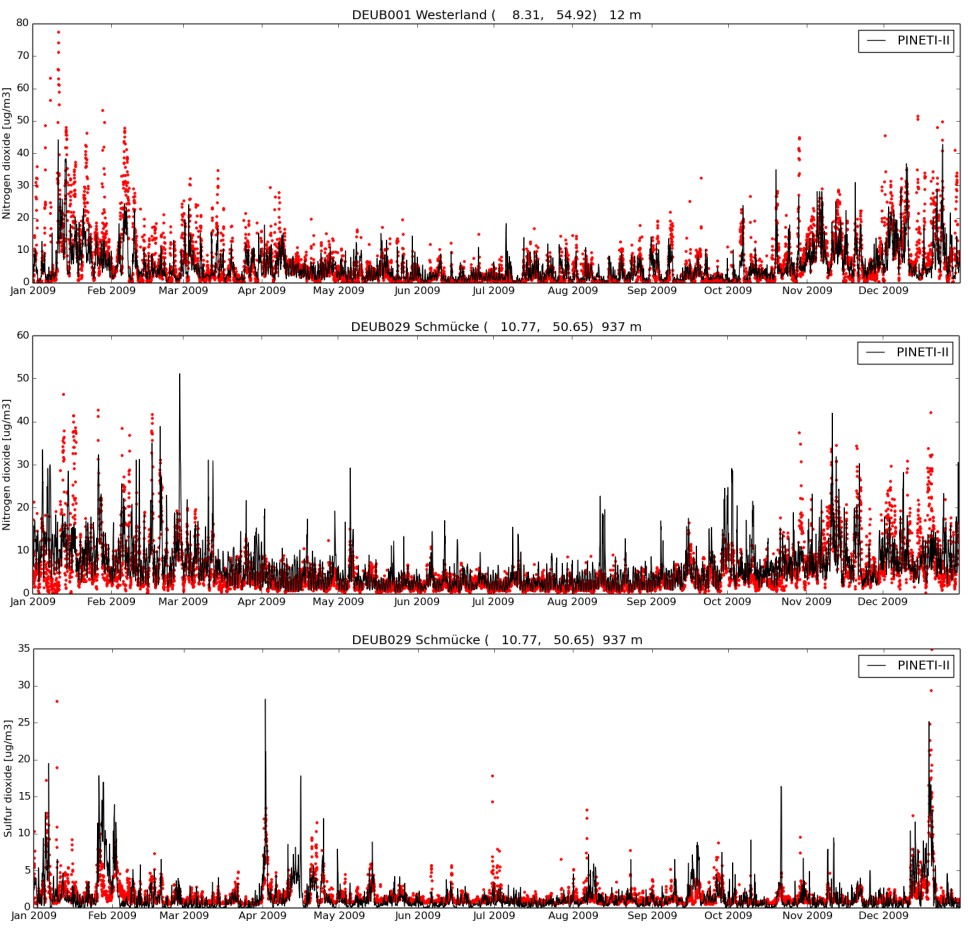






**Figure 6. Comparison between measured and modelled concentration (µg/m³) time series for NO₂ and SO₂ at the UBA stations**
**Westerland and Schmücke.**

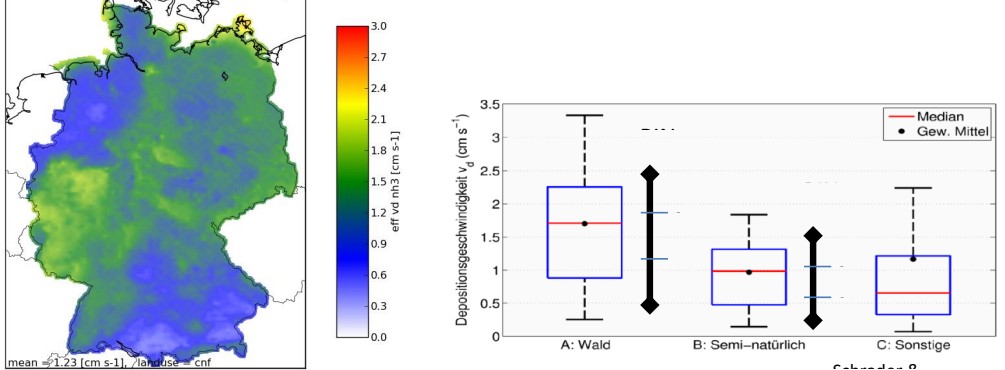


**Figure 7. Dry deposition velocity above coniferous forest (left) and a comparison between the range of annual mean dry deposition**
**velocities for ammonia across Germany and the range average of ammonia deposition velocities reported in literature** (Schrader
and Brümmer, 2014)**.**

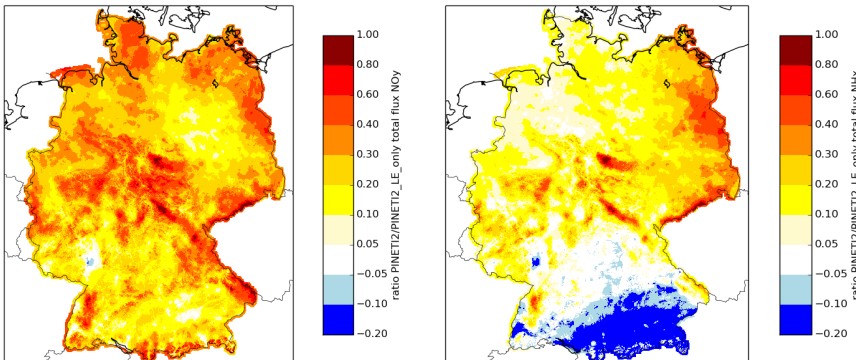


**Figure 8. Relative difference ( (Assessment – LOTOSEUROS) / Assessment) of the total deposition of NOy (left) and NHx (right)**
**between the modelled distributions and the final assessment including empirical wet and occult deposition estimates.**




**Figure 9. Critical load exceedance for reactive nitrogen deposition across Germany**






**Table 1. Enrichment factors for occult deposition as used in this study.**

| Species | Mean enrichment factor |
|---------|------------------------|
| $SO_4^{2-}$ | 7.0 |
| $NO_3^-$ | 8.6 |
| $NH_4^+$ | 9.2 |

**Table 2. Overview of averaged estimates of dry, wet and total deposition fluxes (eq ha$^{-1}$ yr$^{-1}$) per land use category across the**
**German territory for reactive nitrogen. The average over the German territory was obtained using the actual land use distribution**

| Land use | Code | $N_{tot}$ | Dry NHx | Dry NOy | Wet NHx | Wet NOy | Occult NHx | Occult NOy |
|----------|------|-----------|---------|---------|---------|---------|------------|------------|
| Grassland | grs | 901 | 228 | 97 | 327 | 248 | - | - |
| Semi-natural | sem | 948 | 250 | 122 | 327 | 248 | - | - |
| Arable | ara | 982 | 296 | 111 | 327 | 248 | - | - |
| Permanent crops | crp | 1043 | 330 | 137 | 327 | 248 | - | - |
| Coniferous forest | cnf | 1287 | 485 | 182 | 327 | 248 | 26 | 19 |
| Deciduous forest | dec | 1183 | 397 | 162 | 327 | 248 | 28 | 21 |
| Mixed forest | mix | 1235 | 441 | 172 | 327 | 248 | 27 | 20 |
| Water | wat | 861 | 221 | 64 | 327 | 248 | - | - |
| Urban | urb | 1248 | 501 | 172 | 327 | 248 | - | - |
| Other | oth | 894 | 239 | 80 | 327 | 248 | - | - |
| **Germany** | **DEU** | **1057** | **337** | **131** | **327** | **248** | **8** | **6** |


**Table 3. Summary of the statistical model evaluation for SO$_2$ and NO$_2$. The data represent the averages over all N stations. We**
**present the observed and modelled mean concentration as well as the variability expressed as a standard deviation (STD).**
**Furthermore, the bias, root mean squared error (RMSE) and temporal correlation coefficient (COR) are given. The evaluation**
**was performed with time series of daily means.**

| | N | $MEAN_{OBS}$ | $MEAN_{MOD}$ | $STD_{OBS}$ | $STD_{MOD}$ | BIAS | RMSE | $R^2$ |
|---|---|---|---|---|---|---|---|---|
| **SO$_2$** | 31 | 2.1 | 2.4 | 2.0 | 2.3 | 0.26 | 2.2 | 0.59 |
| **NO$_2$** | 45 | 9.0 | 7.4 | 6.5 | 4.8 | -1.5 | 5.1 | 0.71 |


**Table 4. Land use dependent annual effective and average dry deposition velocity at 2.5 meter height (above zero-displacement**
**height and roughness length) across land use types in Germany for six components in cm/s.**

| Vd | NO$_2$ | | NO | | HNO$_3$ | | NH$_3$ | | SO$_2$ | | $NH_4^{2-}$ (fine) | |
|----|--------|---|-----|---|---------|---|--------|---|--------|---|-------------------|---|
| **[cm/s]** | Eff | Ave | Eff | Ave | Eff | Ave | Eff | Ave | Eff | Ave | Eff | Ave |
| **ara** | 0.10 | 0.15 | 0.03 | 0.02 | 1.20 | 1.04 | 0.71 | 0.82 | 0.32 | 0.46 | 0.08 | 0.09 |
| **cnf** | 0.15 | 0.24 | 0.03 | 0.02 | 1.63 | 1.52 | 1.23 | 1.83 | 0.75 | 0.91 | 0.16 | 0.20 |





| | | | | | | | | | | | |
|---|---|---|---|---|---|---|---|---|---|---|---|
| **dec** | 0.13 | 0.21 | 0.03 | 0.02 | 1.63 | 1.52 | 0.95 | 1.48 | 0.74 | 0.90 | 0.16 | 0.20 |
| **grs** | 0.08 | 0.15 | 0.03 | 0.02 | 1.11 | 0.98 | 0.58 | 0.88 | 0.56 | 0.63 | 0.07 | 0.08 |
| **oth** | 0.07 | 0.07 | 0.05 | 0.05 | 0.89 | 0.81 | 0.56 | 0.56 | 0.21 | 0.30 | 0.04 | 0.04 |
| **crp** | 0.13 | 0.18 | 0.03 | 0.02 | 1.42 | 1.22 | 0.81 | 1.03 | 0.66 | 0.76 | 0.10 | 0.11 |
| **sem** | 0.08 | 0.16 | 0.03 | 0.02 | 1.24 | 1.08 | 0.63 | 0.97 | 0.60 | 0.68 | 0.10 | 0.12 |
| **wat** | 0.05 | 0.05 | 0.05 | 0.05 | 0.67 | 0.62 | 0.48 | 0.60 | 0.59 | 0.57 | 0.08 | 0.09 |
| **urb** | 0.08 | 0.08 | 0.08 | 0.08 | 2.94 | 2.58 | 1.09 | 1.32 | 0.43 | 0.82 | 0.14 | 0.17 |
| **mix** | 0.14 | 0.23 | 0.03 | 0.02 | 1.63 | 1.52 | 1.09 | 1.65 | 0.75 | 0.90 | 0.16 | 0.20 |


**Table 5. Comparison of wet deposition fluxes (eq ha⁻¹ yr⁻¹) averaged over all available stations for the year 2009. The bias is**


**provide in an absolute and relative sense**


| Variable | Observed | Modelled | Bias | Relative bias (%) |
|---|---|---|---|---|
| NH$_x$ | 295 | 234 | -61 | -21 |
| NO$_y$ | 226 | 139 | -87 | -38 |

**Table 6. Comparison of the average total NOy, NHx and N deposition for Germany in this study, the LOTOS-EUROS model,**
**EMEP and the previous German assessment MAPESI by Builtjes et al. (2012).**

| | This study | LOTOS-EUROS | EMEP | Builtjes et al., 2011 |
|---|---|---|---|---|
| Year | 2009 | 2009 | 2009 | 2005 |
| NOy | 385 | 298 | 436 | 548 |
| NHx | 672 | 612 | 690 | 895 |
| total N | 1057 | 910 | 1126 | 1443 |


**Table 7. Comparison of the mapped total N- deposition results derived in this study and MAPESI (Builtjes et al., 2011) to**
**empirical derived N deposition estimates (Kg N ha⁻¹ a⁻¹) for three sites across Germany: Forellenbach (Beudert and Breit, 2014),**
**Neuglobsow (Schulte-Bisping and Beese, 2016) and Bourtanger Moor (Mohr, 2013).**

| | Empirical | MAPESI | This study | Ref |
|---|---|---|---|---|
| Forellenbach | 15 | 37 | 19 | Beudert and Breit, 2014 |
| Neuglobsow | 9.5 | 18 | 12 | Schulte-Bisping and Beese, 2016 |
| Bourtanger Moor | 25 (16-35) | 38 | 20 | Mohr, 2013 |
