# Peer review of "Atmospheric nitrogen deposition to terrestrial ecosystems across 2 Germany"

_Biogeosciences, 2017_

## Referee Comment (RC1) · Anonymous Referee #1 · 10 Dec 2017

I will conduct a proper review in due time, but this comment is just to point out that some information is missing:

* Supplementary information. This is referred to in the text, and should contain important background. I cannot find this on the BGD website though.

* References. Many references are to grey literature where the only information given as to source consists of the name of a town! Such references should be completed with the information needed for readers and referees to access these publications. Ideally with web-addresses.

* Of particular importance in this regard the publication of Wichink Kruit et al 2014 is from "Dessau". As this methodology seems important to the methods used in this

paper then it is essential that the authors give a proper reference, and if necessary for the review process they should send a pdf for distribution to the referees,

* There are several other obvious mistakes in the reference list which should be corrected. Authors should proof-read their works and not leave that to referees.

I have one methodological query also for now. Is the Crainwater concentration in equation (1) a simple mean value of measured C, or a precipitation-weighted value as normally used in such estimates?

---

## Referee Comment (RC2) · Anonymous Referee #2 · 9 Jan 2018

This is an interesting article on the nitrogen deposition in Germany with emphasis on some new developments in the methodology for deriving these depositions. In the introduction it is stated that this is indeed the main objective of the manuscript: 'presenting the methodology and illustrating it with results for 2009'. The manuscript claims that compared to previous methodologies some improvements were made: modelling the wet and occult deposition. Although the actual changes to the total deposition are shown in terms of figures/tables, it remains unclear to me to what extend the individual changes played a role in that. For an international audience it might be interesting to know if the improvements presented here also count improve estimates for their countries - but that is not elaborated on here.

Some further remarks are made in the document itself.

[Figure]

Please also note the supplement to this comment:
https://www.biogeosciences-discuss.net/bg-2017-491/bg-2017-491-RC2-supplement.pdf

―――――――――――――――――――

**Supplement:**

[revised manuscript text omitted]

---

## Referee Comment (RC3) · Anonymous Referee #1 · 10 Jan 2018

On December 10th I pointed out that the Supplementary was missing, as was information needed to trace some of the references (the most important being Wichink Kruit et al. 2014, whose methodology seems to be important for this paper). The authors have ignored my comment, which is unacceptable in a discussion forum. As a referee I was expecting this information well within the normal 8 week review process. As it is now, little time remains, and I suggest the paper is rejected for now and re-submitted once all materials are provided.